# Quantum chaos in interacting Bose-Bose mixtures

Tran Duong Anh-Tai[1*], Mathias Mikkelsen[1,2], Thomas Busch[1] and Thomás Fogarty[1†]

**1** Quantum Systems Unit, OIST Graduate University, Onna, Okinawa 904-0495, Japan
**2** Department of Physics, Kindai University, Higashi-Osaka City, Osaka 577-8502, Japan

⋆ tai.tran@oist.jp , † thomas.fogarty@oist.jp

## Abstract

The appearance of chaotic quantum dynamics significantly depends on the symmetry properties of a system, and in cold atomic systems many of these can be experimentally controlled. In this work, we systematically study the emergence of quantum chaos in a minimal system describing one-dimensional harmonically trapped Bose-Bose mixtures by tuning the particle-particle interactions. Using an improved exact diagonalization scheme, we examine the transition from integrability to chaos when the inter-component interaction changes from weak to strong. Our study is based on the analysis of the level spacing distribution and the distribution of the matrix elements of observables in terms of the eigenstate thermalization hypothesis and their dynamics. We show that one can obtain strong signatures of chaos by increasing the inter-component interaction strength and breaking the symmetry of intra-component interactions.

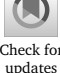

# 1 Introduction

Systems of ultra-cold quantum gases have over the past decades become so highly controllable that they are currently at the forefront of studying the dynamics of quantum few- and many-body physics. In fact, despite the low density they require, it has been experimentally feasible to create strongly-correlated systems of bosons [1–3] and fermions [4–6] in effectively lower dimensional settings. Furthermore, a high degree of control over the external potentials allows to create systems with small, well-defined particle numbers, in which one can explore the cross-over from few- to many-body physics, [4–12]. The stationary properties and dynamics of these few-particle ultra-cold systems have been intensively studied over the last two decades [13,14] and more recently ultra-cold quantum gases were shown to be excellent platforms for experimental studies of equilibration mechanisms in isolated many-body systems since they are almost perfectly decoupled from the environment [15–21].

In particular, numerical calculations and experimental results suggest that for many-body systems which exhibit quantum chaos, unitary non-equilibrium dynamics of an isolated system can lead to the thermalization of observables. One mechanism which ensures thermalization of an observable is the eigenstate thermalization hypothesis (ETH) [22,23] and evidence in a large variety of chaotic systems suggests that their observables obey the ETH [24–27]. One topic in this direction, which has attracted considerable theoretical and experimental interest recently, is how chaotic behavior starts to emerge in small systems consisting of only a couple of particles [28–31], with current cold-atom setups ideally suited to probe these effects. While a standard way to introduce quantum chaos in single-component ultra-cold systems is the breaking of their symmetry properties through controlling the interactions and the external potentials [24–27, 29, 30, 32, 33], ultracold atomic mixtures can also achieve this by using different relative particle numbers or having non-symmetric scattering properties. Despite these additional possibilities, the emergence of quantum chaos in mixtures has only just started to be explored [34], with recent works focusing on systems trapped in double wells [35,36].

While the emergence of chaos in these works is predicated on the geometry of the trapping potential, we show here that the presence of both inter- and intra-component interactions in Bose-Bose mixtures can allow one to observe strongly chaotic dynamics even in a harmonic oscillator. To focus purely on the effect of the interactions, we therefore only consider symmetric binary mixtures which have the same number of particles and all atoms have the same mass. We systematically study the role that intra- and inter-component interactions play in the appearance of chaotic properties of this system using different signatures of quantum chaos such as the spectral statistics, the distribution of the matrix elements of observables with respect to the ETH, and their thermalization dynamics. For this, the manuscript is organized as follows: the Hamiltonian setup and the parameters used for numerical calculations are presented in Sec. 2, and in Sec 3 we show and discuss the key results, including the level spacing distribution, the Brody distribution parameter and the validity of the ETH in detail. Finally, the conclusions are drawn in Sec. 4.

# 2 Model

We consider a bosonic two-component system that is strongly confined in the two transversal directions and can therefore be effectively described by a one-dimensional model. The trapping potential in the longitudinal direction is a harmonic oscillator with frequency $\omega$ and is the same for both components. The masses $m$ of all particles are identical, and the particles interact via a contact interaction modeled by a delta function [37]. Scaling all lengths with the natural oscillator length $a_0 = \sqrt{\hbar/(m\omega)}$ and all energy scales by $\epsilon_0 = \hbar\omega$, the system is

described by the dimensionless Hamiltonian

$$H = \sum_{\sigma \in \{A,B\}} \sum_{i=1}^{N_\sigma} \left[ -\frac{1}{2} \frac{d^2}{d(x_i^\sigma)^2} + \frac{1}{2}(x_i^\sigma)^2 \right] + \sum_{\sigma \in \{A,B\}} g_\sigma \sum_{i<j} \delta(x_i^\sigma - x_j^\sigma) + g_{AB} \sum_{i=1}^{N_A} \sum_{j=1}^{N_B} \delta(x_i^A - x_j^B), \quad (1)$$

where $N_A$ and $N_B$ are the number of particles in the two components $A$ and $B$. In this work, we assume that the number of atoms in each component is identical, $N_A = N_B$, and the symmetry between the two components with respect to all external parameters will help us isolate the influence of the interactions on the system dynamics. The latter are quantified by $g_A$, $g_B$, and $g_{AB}$, which describe the intra-component coupling strengths within the components A and B, and the inter-component coupling strength between the two components, respectively. In principle, all three coupling strengths can be independently varied from the non-interacting ($g = 0$) to the infinitely interacting Tonks-Girardeau (TG) limit ($g \to \infty$) [38] via Feshbach resonances [39].

In the situation with no inter-component interactions, $g_{AB} = 0$, the Hamiltonian separates and each component can be solved independently. Moreover, in a harmonic trap the system is integrable for vanishing intra-component interactions $g_A = g_B = 0$, in the TG limit $g_A = g_B = \infty$ and any combination of these. However, for any finite intra-component interaction, $g_A > 0$ and $g_B > 0$, the harmonic trap breaks integrability, and the system can thermalize, albeit on long timescales owing to the proximity of the system to integrability [40]. The aim of this work is to investigate how the introduction of the inter-species interaction $g_{AB}$ between the two components can further move the system from integrability and elicit strong signatures of chaos.

For $g_{AB} = 0$, exact solutions in terms of single-particle states can be constructed in the non-interacting and TG limit, and exact solutions also exist for the finite interacting two-particle case [41]. However, when the coupling between the components is finite, $g_{AB} > 0$, such solutions are no longer applicable, and the full system requires a numerical treatment. Therefore, to solve Eq. (1), we use an improved numerical exact diagonalization (ED) scheme, the details of which are given in Appendix A. The quantifiers of chaos, such as the level statistics, are sensitive to the symmetry of the states, so for this reason, we focus on bosonic eigenstates with even symmetry, which are subject to both inter- and intra-component interactions. We use an energetically optimized choice of the many-body basis [42, 43], and the effective-interaction approach [44–46] for all three interactions to improve the efficiency of the calculations. While this approach cannot reach the TG limit of infinite interactions, setting the coupling strengths to $g_{\sigma(AB)} = 20$ gives results that are very close to the analytical results. Using this improved ED scheme, accurate results can be obtained for approximately 5400 eigenstates in a system that has two particles in each component (2+2) with a Hilbert space dimension of just $D = 6050$. Furthermore, even though the computations for 3+3 mixtures, which require a Hilbert space of dimension $D = 49460$ in our scheme, are computationally challenging, we are still able to perform calculations for some quantities, which will be discussed in the following section.

## 3 Results and discussion

### 3.1 Level spacing distribution and Brody distribution parameter

As a first step towards exploring the role interactions play in the emergence of quantum chaos, we investigate the level spacing distribution $P(s)$ of the unfolded spectra [24] of the many-body Hamiltonian Eq. (1). Here, $s$ is the spacing between neighboring unfolded eigenvalues. A Poissonian distribution $P_P(s) = \exp(-s)$ is known to correspond to spectra that are uncorrelated and possess degeneracies, which indicates that the system is integrable [27, 47, 48]. On the

other hand, distributions that are of Wigner-Dyson type, $P_{WD} = (0.5\pi s) \exp(-0.25\pi s^2)$, correspond to the spectra that are correlated and non-degenerate, therefore containing avoided crossings. These are well-known signatures of systems that allow for quantum chaos [49]. To avoid isolated energies at the bottom of the spectrum impacting the statistics, we neglect the lowest 5% of eigenstates and only consider the areas of the energy spectrum which possess a high density of states.

Examples of numerically obtained level spacing distributions for different intra- and inter-component coupling strengths are shown in Fig. 1 and compared to the respective Poisson and Wigner-Dyson distributions. In all cases shown, the interaction strength in component A is fixed at $g_A = 10$. Looking at the cases with $g_B \neq g_A$ first (the first and third columns in Fig. 1), one can see that for the 2+2 mixtures (the first two rows), the distribution is close to Poissonian for weak inter-component interactions, $g_{AB} = 1.5$, indicating that the system is close to integrability. For larger inter-component interactions, $g_{AB} = 20$, the distributions match more the Wigner-Dyson shape, with $P(0) \to 0$ indicating significant level repulsion and suggesting that the mixtures are chaotic when the inter-component coupling is strong. However, when the intra-component interactions are equal, $g_B = g_A$, the Wigner-Dyson distribution at strong coupling is lost (panel (e)), level repulsion is reduced, and the tail of the distribution is nearly Poissonian. This suggests that the system is in an intermediate regime owing to the emergence of some degeneracies in the energy spectrum. This behavior is also present in the larger 3+3 system at both weak (panels (i-k)) and strong (panels (l-n)) inter-component interactions. In fact, we see that for the same $g_{AB}$ as the 2+2 case, any indication of Poissonian statistics vanishes already for weak inter-component interactions when the symmetry of the intra-component interactions is broken (panels (i) and (k)). Instead, a significant degree of level repulsion is present, and the distribution is close to Wigner-Dyson.

While visually examining the distributions gives a qualitative picture of the physics, a more quantitative determination of the distribution type can be obtained by fitting it to the Brody distribution

$$P_B(s) = (\beta + 1)bs^\beta \exp(-bs^{\beta+1}). \tag{2}$$

Here $b = \left[\Gamma\left(\frac{\beta+2}{\beta+1}\right)\right]^{\beta+1}$, where $\Gamma(x)$ is the gamma function [24,50]. The fitting parameter $\beta$ is known as the Brody parameter, with $\beta \sim 0$ indicating a Poissonian distribution and $\beta \sim 1$ indicating a Wigner-Dyson distribution. In Figs. 2(a-c) we show $\beta$ for different inter-component coupling strengths $g_{AB} = \{5, 10, 20\}$ as a function of the intra-component coupling strengths $g_A$ and $g_B$. One can immediately see that the off-diagonal regions, corresponding to the situation where the intra-component interactions in the two components are different, $g_A \neq g_B$, have large values of $\beta$, with $\beta \to 1$ with increasing inter-component coupling strengths. However, the Brody parameter takes lower values along the diagonal, i.e. whenever $g_A = g_B$, indicating that the energy spacing distributions are far from Wigner-Dyson. We also note that the continuous transition transversal to the diagonal becomes wider for larger intra-component interactions. The reason for this broadening is that both components are closer to the TG limit, and therefore the spectra do not significantly change even if the interaction strengths are not exactly the same. This effectively widens the condition of equal interaction strength in the two components.

In Fig. 2(d), we show a cut of the panel (b) at $g_A = 10$ and the same data for a 3+3 system. For both system sizes, the Brody parameter drops to a minimum of $\beta \approx 0.3$ at $g_A = g_B = 10$, while away from this point, it is effectively constant with values $\beta \gtrsim 0.8$. This suggests that chaos is not sensitive to the specific values of individual intra-component interactions but that they need to be sufficiently different, $g_A \neq g_B$, and that $g_{AB}$ should be sufficiently large. This can also be seen in panel (e), where we show $\beta$ as a function of $g_{AB}$. One can see that $\beta$ saturates and the Brody distribution is close to Wigner-Dyson when $g_{AB} \gtrsim 2$ for 3+3 and $g_{AB} \gtrsim 6$ for 2+2 systems. Again, the larger system shows signatures of chaos over a wider

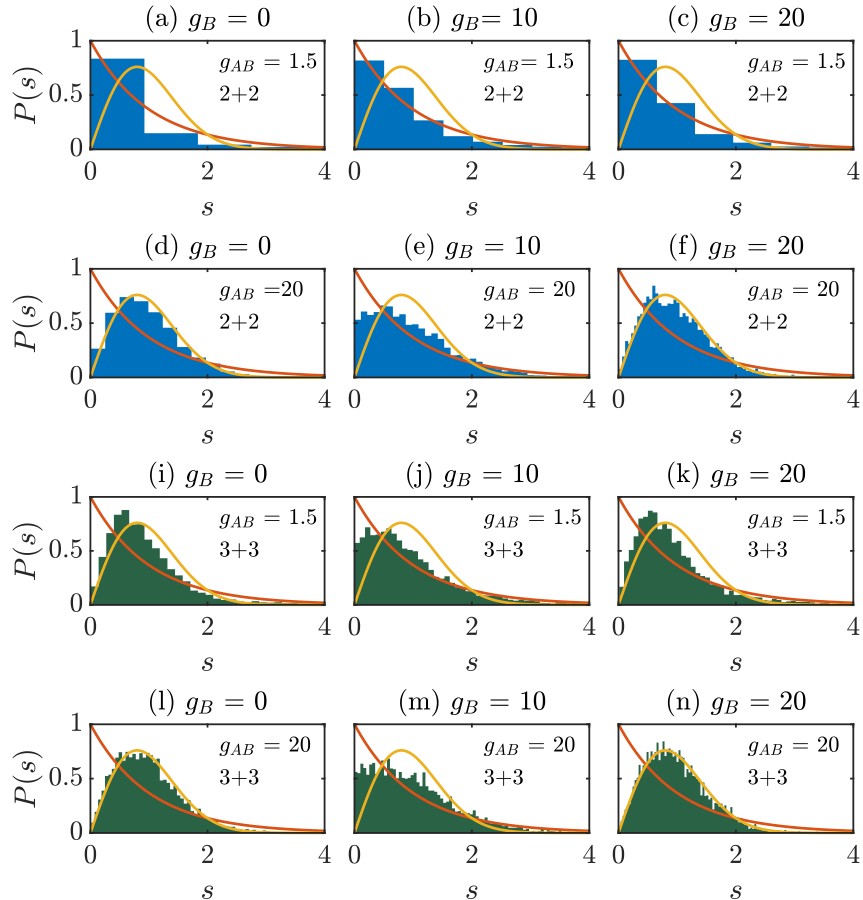

Figure 1: The level spacing distribution $P(s)$ of the unfolded energy spectra for $g_B = 0$ (the first column), $g_B = 10$ (the second column), and $g_B = 20$ (the third column) for various inter-component interaction strengths $g_{AB}$. The intra-component interaction of component A equals to $g_A = 10$ in all panels. The blue bars (a-f) show the results for 2+2 mixtures, while the green bars (i-n) depict them for 3+3 mixtures. The orange and yellow solid lines correspond to the Poissonian and the Wigner-Dyson distribution, respectively.

range of parameters, which is also consistent with the intermediate region around $g_A = g_B$ being narrower in panel (d). Chaos is therefore enhanced for larger systems as the increased density of states introduces more avoided crossings in the energy spectrum [29]. For small $g_{AB}$, the Brody distribution parameter cannot be fitted (when $g_{AB} < 2$ in the 2+2 mixture and $g_{AB} < 0.8$ in the 3+3 mixture) since the distribution has a picket-fence shape. This is due to the large amount of degeneracies in the energy spectra, which emerge as the two components become separable, highlighting that the whole system is close to integrable when $g_{AB} \to 0$ regardless of the choice of $g_A$ and $g_B$.

## 3.2 Dynamics of observables

While we have shown that robust signatures of chaos are evident in the spectral statistics, the long-time dynamics of observables is a useful way to investigate chaos for quantum systems and is more directly comparable to classical notions of chaos. The time-dependent expectation value of a general observable $\hat{O}$ driven by a Hamiltonian with eigenstates and eigenvalues

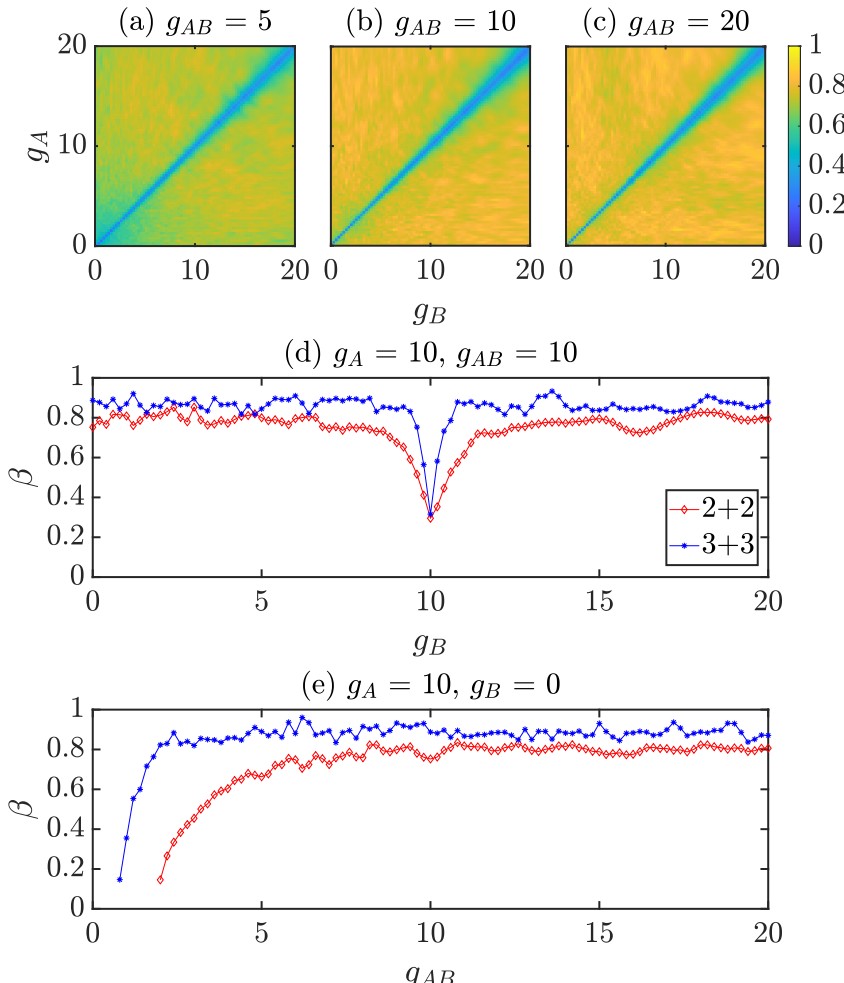

Figure 2: The fitted Brody distribution parameter $\beta$ as a function of $g_A$ and $g_B$ for (a) $g_{AB} = 5$, (b) $g_{AB} = 10$, (c) $g_{AB} = 20$ for 2+2 mixtures. (d) $\beta$ as a function of $g_B$ for $g_A = g_{AB} = 10$, and (e) $\beta$ and as a function of $g_{AB}$ for $g_A = 10$ and $g_B = 0$. Data are shown for both the 2+2 (red diamonds) and 3+3 (blue asterisks) mixtures.

$\{|m\rangle, E_m\}$ is given by

$$O(t) = \langle\Psi(0)|e^{i\hat{H}t}\hat{O}e^{-i\hat{H}t}|\Psi(0)\rangle = \sum_{m\neq n} c_m^* c_n e^{-i(E_n - E_m)t} O_{mn} + \sum_m |c_m|^2 O_{mm}. \tag{3}$$

Here $c_m = \langle m|\Psi(0)\rangle$ are the overlaps between the eigenstates $|m\rangle$ and the initial state $|\Psi(0)\rangle$, while $O_{mn} = \langle m|\hat{O}|n\rangle$ are the expectation values of the observable $\hat{O}$ with respect to the eigenstates $|m\rangle$ and $|n\rangle$. To say that an isolated quantum system is thermalized, two conditions must be met. First, after long-time dynamics, the system should relax to a stationary state in which the expectation value of the observable only slightly fluctuates around the infinite-time average, which for non-degenerate systems is given by

$$\lim_{t\to\infty}\left(\frac{1}{t}\int_0^t O(t')dt'\right) = \sum_m |c_m|^2 O_{mm} = O_{\text{DE}}. \tag{4}$$

Since the infinite-time average depends only on the diagonal terms in Eq. (3), this quantity is often called the diagonal ensemble (DE). The second condition is that the DE average equals

the microcanonical ensemble (ME) average

$$O_{\text{DE}} = O_{\text{ME}} \,, \tag{5}$$

where

$$O_{\text{ME}} = \frac{1}{N_{mc}} \sum_m O_{mm} \,. \tag{6}$$

In the definition of the ME average, $N_{mc}$ is the number of eigenstates with energies $E_m$ satisfying $|E_m - E_{\text{mid}}| \leq \Delta E$ where $E_{\text{mid}}$ and $\Delta E$ are the center and the width of the energy window, respectively.

It can be shown that these two conditions can be enforced by two constraints, commonly referred to as the eigenstate thermalization hypothesis (ETH) [22, 23, 51–53], on the matrix elements of the operator $O_{mn}$ with respect to the Hamiltonian eigenstates. The first constraint is on the diagonal elements, namely that $O_{mm}$ is a smooth function of $m$ which ensures that Eq. (5) is fulfilled [47, 54–56]. The second constraint is on the off-diagonal elements, which determine the time-dependent fluctuations as can be seen from Eq. (3). It can be shown that these fluctuations will decay to zero with the system size if the distribution of off-diagonal elements is Gaussian [29, 57–59], which is therefore the second constraint. Chaotic systems are generally expected to obey the ETH and thermalize, and in the following sections, we will investigate some representative observables for 2+2 mixtures. We will first check the distribution of their off-diagonal matrix elements and, if these follow the off-diagonal ETH, use this as an indicator for quantum chaos. In the next step, we will also look at the long-time dynamics and the appearance of thermalization in the identified chaotic regimes.

### 3.2.1 Off-diagonal matrix elements

It is common to determine the Gaussianity of the distribution of off-diagonal elements $O_{mn}$ via the kurtosis [57–60], which is defined as

$$\mathcal{K}_{\hat{O}} = \frac{\langle (O_{mn} - \langle O_{mn} \rangle)^4 \rangle}{\text{std}(O_{mn})^4} \,, \tag{7}$$

where $\text{std}(X)$ is the standard deviation of the distribution and $\langle \cdot \rangle$ is the average over all eigenstates $|m\rangle \neq |n\rangle$. The kurtosis of a Gaussian distribution is precisely 3, which is, therefore, a well-defined indicator of quantum chaos, whereas, in integrable systems, it can take much larger values.

As our observable we choose the trapping potential operator of the entire system

$$\hat{U} = \frac{1}{2} \sum_{\sigma \in \{A,B\}} \sum_{i=1}^{N_\sigma} (x_i^\sigma)^2 \,, \tag{8}$$

and choose a fixed number of eigenstates high in the spectrum to compute the off-diagonal terms $U_{mn} = \langle m|\hat{U}|n\rangle$. To emphasize the chaotic regime we plot the inverse of the kurtosis, $\mathcal{K}_{\hat{O}}^{-1}$, in Figs. 3(a-c) for the case of the 2+2 system. The inverse kurtosis increases as the inter-component coupling strengths are increased, attaining values close to 1/3 when $g_A \neq g_B$ indicating that the system is chaotic. Additionally, the inverse kurtosis takes noticeably lower values along the diagonal where $g_A = g_B$, a cut of which is shown in panel (d) for 2+2 (red diamonds) and 3+3 mixtures (blue asterisks). In Fig. 3(e), we also show the dependence of the inverse kurtosis on the inter-component interaction for fixed $g_A = 10$ and $g_B = 0$. In the 2+2 system, the inverse kurtosis saturates to its maximum value for interactions of $g_{AB} \gtrsim 6$, whereas for the 3+3 system, the inverse kurtosis almost reaches 1/3 when $g_{AB} \gtrsim 2.5$. For the larger system, it is obvious that the inverse kurtosis is much closer to 1/3 when the

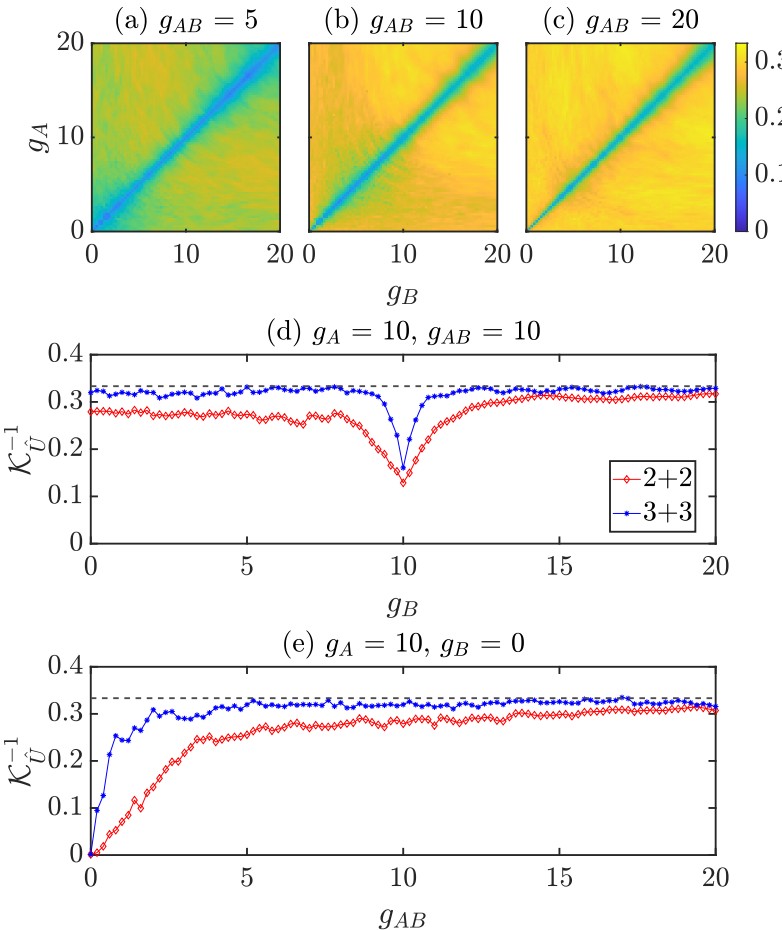

Figure 3: The inverse kurtosis, $\mathcal{K}_{\hat{U}}^{-1}$, of the distribution of the off-diagonal elements $U_{mn}$ as a function of the intra-component coupling strengths for (a) $g_{AB} = 5$, (b) $g_{AB} = 10$, (c) $g_{AB} = 20$ for 2+2 systems. Panel (d) shows the dependence of the inverse kurtosis on the intra-component coupling strength $g_B$ for $g_A = g_{AB} = 10$, while panel (e) depicts that of the inverse kurtosis on the inter-component coupling strength for $g_A = 10$, and $g_B = 0$ for the 2+2 and 3+3 mixtures. Note that the maximum value of $\mathcal{K}^{-1}$ is 1/3 in the chaotic limits, as shown by the black dashed line in panels (d-e), and its minimum value is 0 in the integrable limit. In the calculations, we choose a set of 201 eigenstates in the range of 2600-2800 (3800-4000) from a total of 6050 (49460) eigenstates for 2+2 (3+3) mixtures.

inter-component interaction is strong, indicating a noticeably higher level of chaos than the 2+2 system. The inverse kurtosis is remarkably similar to the results of the spectral analysis depicted in Fig. 2, re-enforcing that strong inter-species interactions ($g_{AB} \gg 0$) and unequal intra-species interactions ($g_A \neq g_B$) are indeed necessary for strong chaos. We also note that in contrast to interacting two-component systems in free space [61], the trapped system we consider does not contain an integrable point at $g_A = g_B = g_{AB}$. While the system does possess SU(2) symmetry at this point, there is nothing evident in the energy spacing statistics or the kurtosis that differentiates it from any other point along the diagonal $g_A = g_B$.

For a more in-depth look at the chaos-integrability transition as a function of the intra-component interactions, we show the distribution of the off-diagonal elements $\hat{U}_{mn}$ as a function of $g_B$ for $g_A = g_{AB} = 10$ in Fig. 4(a). The distribution away from $g_B = g_A$ is uniform and unaffected by the specific choice of $g_B$, whereas close to $g_A = g_B$ the shape of the distribu-

tion changes and acquires a sharp peak. We show three slices of this region of the parameter space in Figs. 4(b-c), which correspond to the situations far from, close to, and at the transition point, $g_B = g_A$. For $g_B = 6$ (Fig. 4(b)), the distribution is well-fitted by a Gaussian (red solid line), which is a strong indicator of chaos. For $g_B = 9.8$, which is close to the transition point, the distribution loses the Gaussian shape as shown in Fig. 4(c). At the transition point $g_B = g_A = 10$ (Fig. 4(d)), a sharp peak with large probability at $U_{mn} = 0$ and Gaussian tails is observed, which therefore leads to a reduction in the inverse kurtosis.

The origin of this peak can be understood from the magnitude of the off-diagonal elements $|U_{mn}|$, shown in Figs. 4(f-i) as a function of the energy difference $\omega_{mn} = |E_m - E_n|$ between the $|m\rangle$ and $|n\rangle$ states. Away from $g_A = g_B$ (Fig. 4(f) and (g)), all elements are finite; however, at $g_A = g_B = 10$ the elements separate into two different bands. The upper band lies in the same range as the elements in panels (f) and (g), while the lower band takes infinitesimal values and is responsible for the peak in panel (d). We find that the elements that vanish consist of eigenstates $|m\rangle$ that do not contain the same sub-component Fock states, i.e., $|200\ldots\rangle \otimes |200\ldots\rangle$, $|101\ldots\rangle \otimes |101\ldots\rangle,\ldots$ etc. This suggests the emergence of a new symmetry sector when the interactions are equal, $g_A = g_B$, whose eigenstates are decoupled from some states with bosonic symmetry. When we remove these eigenstates (accounting for nearly 50% of the whole spectrum) and recompute the distribution of the off-diagonal elements $U_{mn}$, the sharp peak at $U_{mn} = 0$ is eliminated since all elements are now finite as shown in Fig. 4(i) and the Gaussian distribution is recovered (see Fig. 4(e)). Overall we can conclude that the two-component bosonic system confined harmonically is not fully chaotic in the strong inter-component interacting regime when the intra-component coupling strengths are identical, which is consistent with the spectral analysis. Finally, we note that even slightly breaking the symmetry of the interactions, as shown in panel (c) with $g_B = 9.8$, also results in a reduction of chaos; however, no extra symmetry sector is visible in panel (h) as all combinations of eigenstates have a finite $U_{mn}$.

### 3.2.2 Dynamics and thermalization

Let us next evaluate the nonequilibrium dynamics induced by Hamiltonian Eq. (1) to further assess the presence of chaos identified in the previous sections. To do this, we need to identify an appropriate initial state. A minimum requirement is that the initial state is not an eigenstate of the Hamiltonian, as this would lead to trivial dynamics. In order to investigate the thermalization properties and compare them with the ME, the energy of the initial state should be far from the bottom of the spectrum. A good choice of the initial state which fulfills these requirements is the product state $|\Psi(0)\rangle = |\Psi^A(0)\rangle \otimes |\Psi^B(0)\rangle$ where

$$
\begin{aligned}
|\Psi^A(0)\rangle = \mathcal{A} \sum_{\substack{i,j=1 \\ i\neq j}}^{2} \Big[ &\phi_0(x_i^A)\phi_{19}(x_j^A) + \phi_1(x_i^A)\phi_{18}(x_j^A) \\
&+ \phi_2(x_i^A)\phi_{17}(x_j^A) + \phi_3(x_i^A)\phi_{16}(x_j^A) \\
&+ \phi_4(x_i^A)\phi_{15}(x_j^A) + \phi_5(x_i^A)\phi_{14}(x_j^A) \\
&+ \phi_6(x_i^A)\phi_{13}(x_j^A) + \phi_7(x_i^A)\phi_{12}(x_j^A) \\
&+ \phi_8(x_i^A)\phi_{11}(x_j^A) + \phi_9(x_i^A)\phi_{10}(x_j^A) \Big],
\end{aligned}
\tag{9}
$$

$$
|\Psi^B(0)\rangle = \mathcal{B} \sum_{\substack{i,j=1 \\ i\neq j}}^{2} \Big[ \phi_0(x_i^B)\phi_0(x_j^B) + \phi_0(x_i^B)\phi_1(x_j^B) \Big].
\tag{10}
$$

Here $\phi_n(x)$ denotes the $n$-th single-particle eigenfunction of the non-interacting 1D harmonic oscillator and $\mathcal{A}$ and $\mathcal{B}$ are normalization constants. The energy of the initial state, with



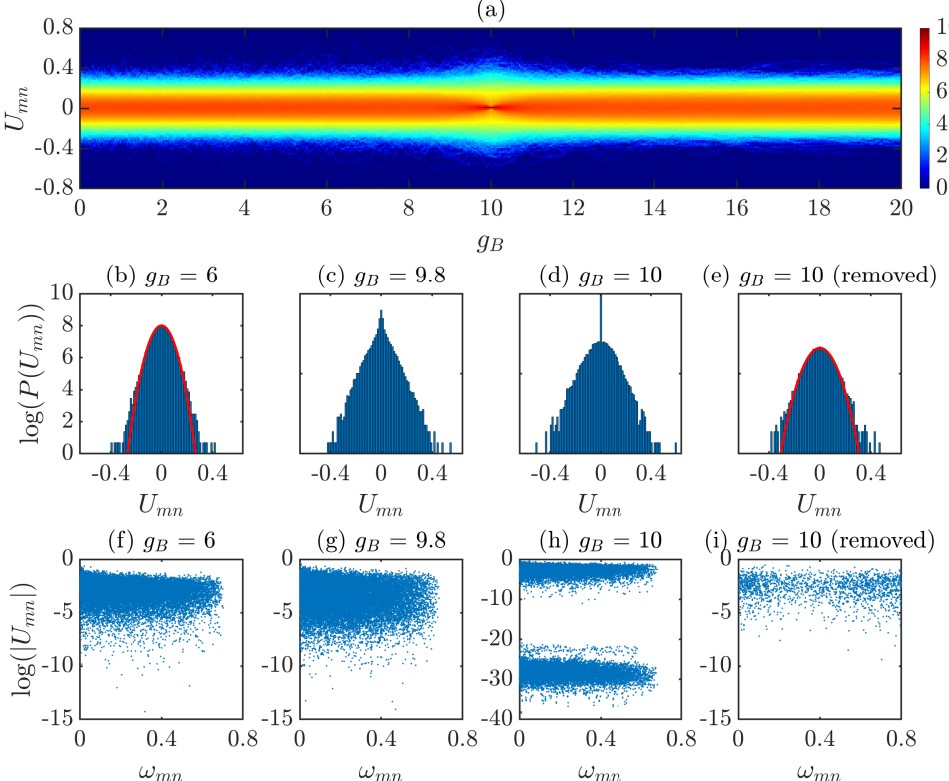

Figure 4: (a) Logarithm of the distribution for the off-diagonal elements $\hat{U}_{mn}$ as a function of $g_B$ for $g_A = g_{AB} = 10$. (b-d) Three slices at $g_B = 6, 9.8, 10$ of the distribution of the off-diagonal elements $\hat{U}_{mn}$ and (f-h) their magnitude $|\hat{U}_{mn}|$ as a function of $\omega_{mn} = |E_m - E_n|$. Panels (e) and (i) are the same as panels (d) and (h), respectively, after removing the eigenstates that do not include the identical sub-component Fock states, i.e., $|200\ldots\rangle \otimes |200\ldots\rangle$, $|101\ldots\rangle \otimes |101\ldots\rangle \ldots$ etc. Note that the solid red line in panels (b) and (e) represents the fitted Gaussian distribution.

respect to the non-interacting Hamiltonian, is $E_{ini} = 21.5$, which is far from the bottom of the spectrum.

We consider the dynamics of this state after a sudden quench of the interaction terms in Eq. (1), with the time-evolved state given by

$$|\Psi(t)\rangle = \exp(-i\widehat{H}t)|\Psi(0)\rangle = \sum_m c_m \exp(-iE_m t)|m\rangle, \qquad (11)$$

where $c_m = \langle m|\Psi(0)\rangle$ denote the overlaps between the initial state $|\Psi(0)\rangle$ and the eigenvectors $|m\rangle$ with energies $E_m$ of the Hamiltonian (1) obtained by the improved exact diagonalization scheme. To calculate the microcanonical ensemble average, the energy window is centered at the expectation value of the Hamiltonian with respect to the initial state

$$E_{\mathrm{mid}} = \sum_m |c_m|^2 E_m, \qquad (12)$$

and the width of the window, $\Delta E = 2$, is chosen such that the number of eigenstates in the window is on the order of $10^3$ and is sufficient for our results to converge. In the following, we focus on the 2+2 system, and to probe the thermalization dynamics, we compute the difference between the dynamical trapping potential energy and its microcanonical ensemble average $U_{\mathrm{ME}}$

$$\Delta_U(t) = U(t) - U_{\mathrm{ME}}. \qquad (13)$$

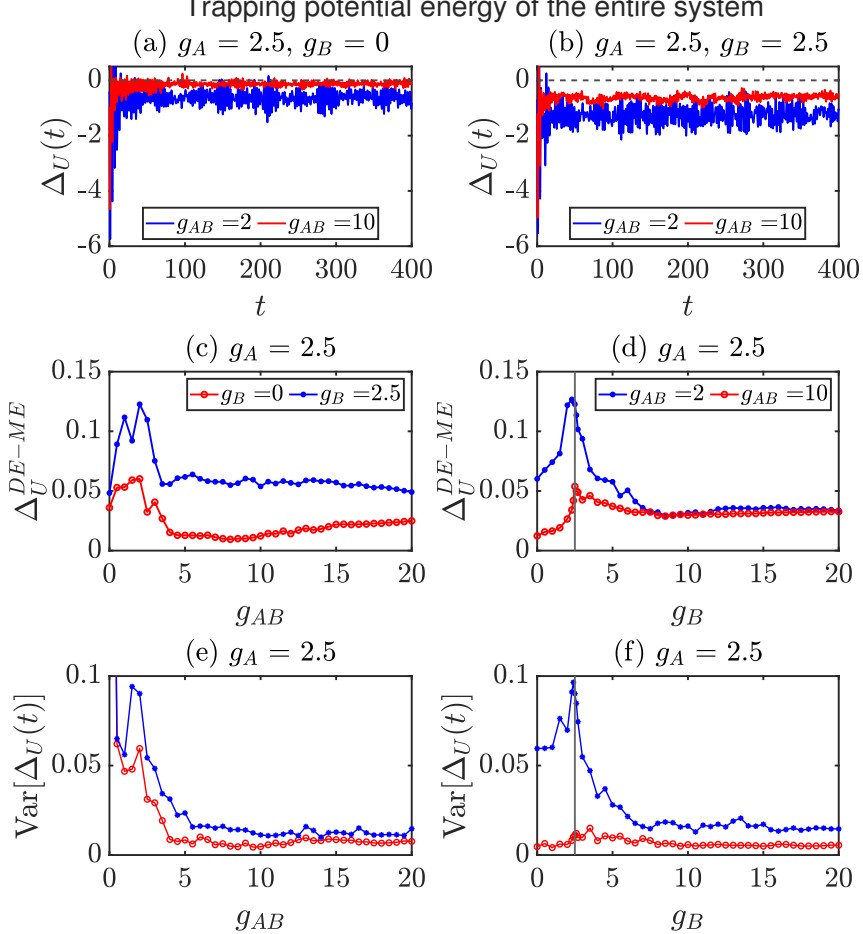

Figure 5: (a-b) Time evolution of $\Delta_{\hat{U}}(t)$ for $g_{AB} = 2$ (all blue lines) and $g_{AB} = 10$ (all red lines) for $g_B = \{0, 2.5\}$. (c-d) The relative deviation between the diagonal and microcanonical ensembles $\Delta^{DE-ME}$ as a function of (c) $g_{AB}$ with $g_B = 0$ (red line) and $g_B = 2.5$ (blue line), and (d) $\Delta^{DE-ME}$ as a function of $g_B$ with $g_{AB} = 2$ (blue line) and $g_{AB} = 10$ (red line). (e-f) The variance of $\Delta_U(t)$ as a function of (e) $g_{AB}$ and (f) $g_B$ with the same parameters as (c-d). In all cases, the intra-component interaction of component A is fixed at $g_A = 2.5$. The initial state is chosen as equation (9), and the energy window is centered at the expectation energy of the quench with the width $\Delta E = 2$. For illustrative purposes, the horizontal black dashed line in panels (a) and (b) depict $\Delta_U(t) = 0$, while the vertical black lines in panels (d) and (f) highlight the point where $g_A = g_B = 2.5$.

In Fig. 5(a,b), we show the time evolution of $\Delta_U(t)$ with the above-mentioned initial state and energy window for the case of equal intra-component interactions, $g_A = g_B = 2.5$, and the unequal case with $g_A = 2.5$ and $g_B = 0$. In both cases, when the inter-component interaction is weak, $g_{AB} = 2$ (blue lines), $\Delta_U(t)$ has large oscillations and is far from zero during the long timescales we consider. This shows that the dynamical trapping potential energies do not relax to the microcanonical ensemble average value when the inter-component interaction is weak. On the contrary, when the inter-component interactions are strong, $g_{AB} = 10$ (red lines), $\Delta_U(t)$ quickly relaxes to its infinite time average and only possesses small oscillations about its mean. When the intra-component interactions are different, $g_A = 2.5$ and $g_B = 0$, it is evident from Fig. 5(a) that $\Delta_U(t)$ fluctuates around zero, showing that the diagonal ensemble is close to the microcanonical ensemble and thus the dynamics can be said to be thermalized.

Meanwhile, $\Delta_U(t)$ for equal intra-component interactions, $g_A = g_B = 2.5$ relaxes to a value different from zero as can be seen clearly in Fig. 5(b), and therefore does not thermalize.

A more quantitative understanding of how closely the system approaches the microcanonical ensemble prediction can be gained by evaluating its relative difference from the diagonal ensemble [22]

$$\Delta_U^{\text{DE}-\text{ME}} = \left| \frac{U_{\text{DE}} - U_{\text{ME}}}{U_{\text{DE}}} \right| . \tag{14}$$

In Fig. 5(c) we show this as a function of $g_{AB}$ for fixed $g_A = 2.5$ and for $g_B = \{0, 2.5\}$. As seen in Fig. 5(c), when the inter-component interactions are large, $g_{AB} \gtrsim 4$, the relative deviations between the two ensembles are small for $g_B = 0$ showing the concrete signatures of quantum chaos in this regime. On the contrary, for $g_B = g_A = 2.5$, the difference $\Delta_U^{\text{DE}-\text{ME}}$ takes comparatively larger values, indicating that the system is in the intermediate regime between chaos and integrability. For weak inter-component interactions, $g_{AB} < 4$, the system should not be chaotic, and indeed $\Delta_U^{\text{DE}-\text{ME}}$ takes larger values for both equal $g_A = g_B$ and unequal $g_A \neq g_B$ intra-component interactions, with the symmetric interacting system always being further from thermalization. However, at $g_{AB} = 0$, these relative deviations both take comparable and small finite values, in apparent contradiction to the results from the previous sections where any signatures of chaos should vanish.

For a more quantitative understanding of the relaxation process at small $g_{AB}$, we examine the variance of $\Delta_U(t)$, $\text{Var}[\Delta_U(t)]$. We consider the variance in the time period between $t = 100$ and $t = 400$, which is chosen to avoid the large oscillations in short-time scales just after the quench while still capturing small fluctuations at long-time scales. In Fig. 5(e), we show $\text{Var}[\Delta_U(t)]$ as a function of $g_{AB}$ for the two cases in Fig. 5(b). In all situations, $\text{Var}[\Delta_U(t)]$ takes large values at $g_{AB} = 0$ (on the order of 1 which is out of the scale in the figure), indicating excessive oscillations about the diagonal ensemble value and the absence of equilibration. While for $g_{AB} \gtrsim 4$, the variance declines from large values to effectively zero, showing that the system equilibrates in both cases. In addition to the Brody distribution parameter $\beta$ and the kurtosis of the off-diagonal distribution $\mathcal{K}_{\hat{U}}$, the crossover between chaos and integrability occurring when $g_A = g_B$ is also captured by $\Delta_U^{\text{DE}-\text{ME}}$ and $\text{Var}[\Delta_U(t)]$ as shown by the sharp peak in Fig. 5(d) and (f), respectively. The strong inter-component interaction regime $g_{AB} = 10$ is generally shown to relax closer to the microcanonical ensemble than for the weaker interactions $g_{AB} = 2$, but for $g_B > 7$ both cases converge to the same value (see panel (d)). However, the variance again indicates that strong inter-component interactions ($g_{AB} = 10$) are necessary for equilibration, as fluctuations in the potential energy essentially vanish for any $g_B$ (see panel (f)).

In the following, we investigate the density distribution of component B, which is an experimentally accessible observable. The initial density distributions of components A and B are depicted in Fig. 6(a), with component B offset from the center of the trap while the density of component A is symmetric about the trap center. Similar to the trapping potential energy, we also probe the absolute difference between the dynamical density distribution function of component B and its microcanonical ensemble average

$$\delta_{n^B}(x^B, t) = \left| n^B(x^B, t) - n_{ME}^B(x^B) \right| . \tag{15}$$

To summarize this information in a single number, independent of the position $x$, we also consider the integration of $\delta_{n^B}(x^B, t)$ over position

$$\Delta_{n^B}(t) = \int \delta_{n^B}(x^B, t) dx^B . \tag{16}$$

If $\delta_{n^B}(x^B, t)$ and $\Delta_{n^B}(t)$ vanish on long-time scales, this clearly demonstrates that the system has thermalized.

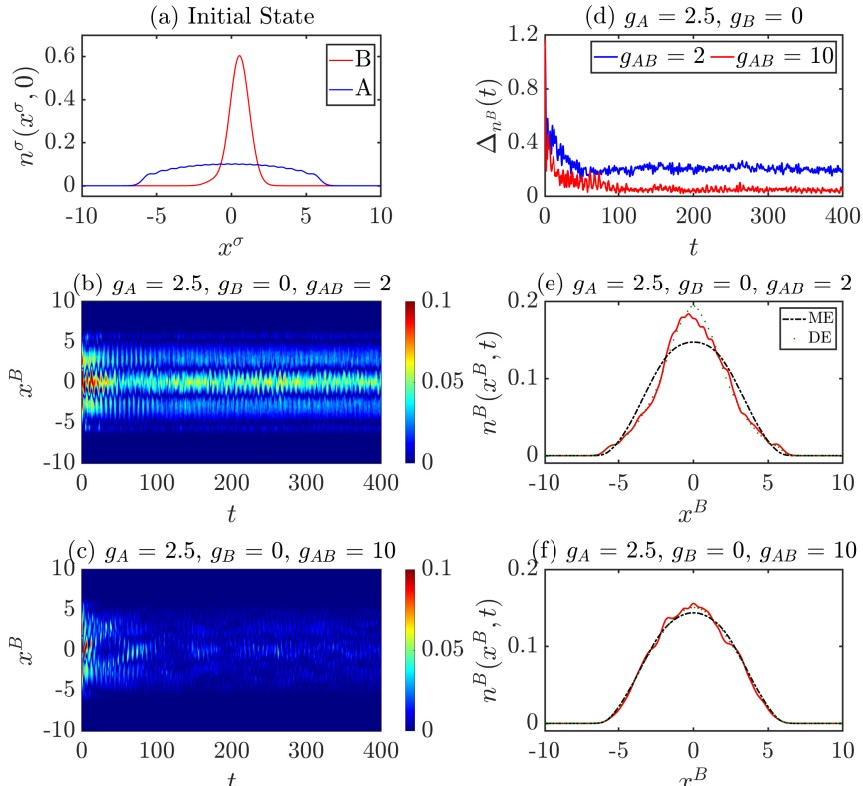

Figure 6: (a) The density distribution functions of the initial state given by Eq. (9). (b-c) The absolute difference between the dynamical density distribution function of component B and its ME average $\delta_{n^B}(x^B, t)$ (see Eq. (15)). (d) The absolute difference between the dynamical density distribution function of component B and its microcanonical average integrated over the position space $\Delta_{n^B}(t)$ (see Eq. (16)). (e-f) Comparison of the density $n^B(x^B, t)$ at $t = 200$ (red line) with its average predicted by the microcanonical (black dash line) and diagonal (green dots) ensembles.

In Fig. 6(b) we show the dynamics of $\delta_{n^B}(x^B, t)$ outside the chaotic regime for $g_A = 2.5$, $g_B = 0$ and $g_{AB} = 2$, while in Fig. 6(e) we show a snapshot of the density at $t = 200$ and compare it to the microcanonical and diagonal ensembles. The density does not relax to the ME prediction on the timescales we consider and retains a higher probability in the middle of the trap than the ME density during the whole dynamics. In comparison, the dynamics in the chaotic regime with $g_{AB} = 10$ shows that $\delta_{n^B}(x^B, t)$ relaxes to the ME after $t \approx 100$ (see panels (c) and (f)). Finally, in Fig. 6 (d) we show the integrated density difference $\Delta_{n^B}(t)$ for the two regimes, affirming that while both systems do equilibrate, strong inter-species interactions ($g_{AB} \gg 1$) are required for thermalization. In addition, we note that the thermalization time of the dynamics shown in Figs. 5 and 6 is approximately two orders of magnitude larger than the two-body collision time [62–64].

Finally, we investigate the difference between the DE and ME predictions of the integrated density distribution for component B,

$$\Delta_{n^B}^{\text{DE}-\text{ME}} = \int \left| n^B_{\text{DE}}(x^B) - n^B_{\text{ME}}(x^B) \right| dx^B \,. \tag{17}$$

Fig. 7(a) shows $\Delta_{n^B}^{\text{DE}-\text{ME}}$ as a function of the inter-component interaction strength $g_{AB}$ for $g_A = 2.5$ and $g_B = \{0, 2.5\}$. For $g_{AB} \leq 3$, the deviation between the two ensembles $\Delta_{n^B}^{\text{DE}-\text{ME}}$ remains large, indicating that the long-time average is not described by the ME. We also point

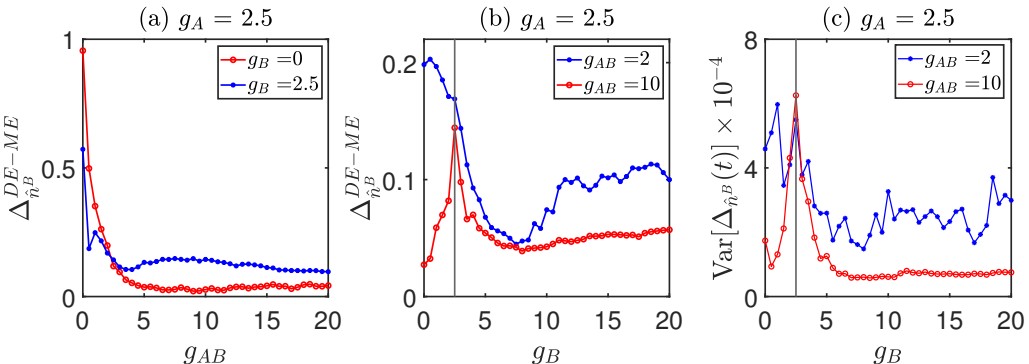

Figure 7: (a-b) The deviation $\Delta_{n^B}^{DE-ME}$ as a function of $g_{AB}$ (panel (a)) and $g_B$ (panel (b)). (c) The variance of $\Delta_{n^B}(t)$ as a function of $g_B$. In all cases, the intra-component interaction of component A is fixed at $g_A = 2.5$. The initial state is chosen as equation (9), and the energy window is centered at the expectation energy of the quench with the width $\Delta E = 2$. The vertical black lines in panels (b) and (c) depict the point where $g_A = g_B = 2.5$.

out that the density is unambiguous in this regard, with $\Delta_{n^B}^{\mathrm{DE-ME}}$ attaining its maximum for $g_{AB} = 0$ as expected, which is in contrast to the potential energy in Fig. 5. For larger inter-species interactions, $g_{AB} > 3$, the difference between the ensembles has a similar trend to the potential energy, with the system with unequal intra-species interactions $g_A \neq g_B$ eliciting stronger signatures of chaos. We can see this also in Fig. 7(b,c) where we consider $\Delta_{n^B}^{\mathrm{DE-ME}}$ and the variance $\mathrm{Var}[\Delta_{n^B}(t)]$, as a function of $g_B$. Strong indications of thermalization are present when $g_A \neq g_B$ and $g_{AB} = 10$, with both $\Delta_{n^B}^{\mathrm{DE-ME}}$ and $\mathrm{Var}[\Delta_{n^B}(t)]$ possessing small values, with the DE having around a 5% deviance from the ME. In general, there are stronger signatures of chaos in the density for $g_{AB} = 10$ rather than $g_{AB} = 2$, with the data being qualitatively similar to the potential energy in Fig. 5. Overall the results confirm the chaos-integrability transitions of binary mixtures with respect to the intra- and inter-component interactions predicted by the spectral statistics and are robust evidence indicating that strong inter-component interactions and unequal intra-component interactions are the root cause of quantum chaos in these mixtures.

## 4 Conclusion

To summarize, we have numerically investigated the emergence of quantum chaos and integrability in a binary bosonic mixture in terms of their intra- and inter-component interactions. Exploring the spectral statistics, including the level spacing distribution and the Brody parameter, we have found that quantum chaos can be caused by strong inter-component interactions and the breaking of the symmetry of the intra-component interactions. Interestingly, we have also observed that although the degree of quantum chaos increases with the growth of the inter-component coupling strength, the system remains non-chaotic whenever the intra-component interactions in the two components are identical. The results from the spectral statistics were confirmed by examining the distribution of matrix elements of the trapping potential energy, producing remarkably similar results. In addition, we have validated the central concept of the ETH, which states that the existence of quantum chaos guarantees the system will thermalize once it is driven out of equilibrium by analyzing the real-time dynamics of some observables. Additionally, we see evidence that the system becomes more chaotic when the particle number is increased, i.e., the system with six bosons, $N_A = N_B = 3$, is more chaotic

than that with four bosons, $N_A = N_B = 2$, as depicted in Fig. 2(d-e) and Fig. 3(d-e). This suggests that strong signatures of chaos can already emerge in small quantum systems as the 3+3 system already saturates our chaos indicators, adding further evidence to recent works in this direction [28–31, 36]. Future extensions to this work could focus on enhancing chaos by breaking further symmetries in the system, for instance, by considering both mass and particle imbalanced mixtures along with Bose-Fermi systems.

## Acknowledgement

The authors thank Miguel Ángel García-March for insightful discussions.

**Funding information** TF acknowledges support from JSPS KAKENHI Grant Number JP23K03290. TF and TB are also supported by JST Grant Number JPMJPF2221. MM is supported by JSPS KAKENHI Grant Number JP22K1400. The authors gratefully acknowledge the help and financial support of the Okinawa Institute of Science and Technology Graduate School (OIST), including the high-performance computing resources provided by the Scientific Computing and Data Analysis section at OIST.

## A  Improved exact diagonalization scheme

The numerical approach employed in this paper uses the second quantization formalism to treat the many-body problem. For this we introduce the time-independent bosonic field operators $\widehat{\Psi}_\sigma(x)$ and $\widehat{\Psi}_\sigma^\dagger(x)$ that destroys and creates a boson of type $\sigma \in \{A, B\}$ at the position $x$, respectively, and which satisfy the commutation relations

$$\left[\widehat{\Psi}_\sigma(x), \widehat{\Psi}_{\sigma'}^\dagger(x')\right] = \delta_{\sigma\sigma'}\delta(x - x'), \tag{A.1}$$

$$\left[\widehat{\Psi}_\sigma^\dagger(x), \widehat{\Psi}_{\sigma'}^\dagger(x')\right] = \left[\widehat{\Psi}_\sigma(x), \widehat{\Psi}_{\sigma'}(x')\right] = 0. \tag{A.2}$$

To numerically represent the system, it is useful to expand the field operators in terms of a complete discrete set of single-particle (SP) basis states $|\phi_{\sigma,i}\rangle$ (with associated wavefunctions $\phi_{\sigma,i}(x) = \langle x|\phi_{\sigma,i}\rangle$) as

$$\widehat{\Psi}_\sigma(x) = \sum_i \phi_{\sigma,i}(x)\widehat{a}_{\sigma,i}, \tag{A.3}$$

$$\widehat{\Psi}_\sigma^\dagger(x) = \sum_i \phi_{\sigma,i}^*(x)\widehat{a}_{\sigma,i}^\dagger. \tag{A.4}$$

Here $\widehat{a}_{\sigma,i}^\dagger$ and $\widehat{a}_{\sigma,i}$ create/destroy a particle of component $\sigma$ in the single-particle state $|\phi_i\rangle$ and obey the bosonic commutation relations

$$\left[\widehat{a}_{\sigma,j}, \widehat{a}_{\sigma',k}^\dagger\right] = \delta_{\sigma\sigma'}\delta_{jk}, \tag{A.5}$$

$$\left[\widehat{a}_{\sigma,j}^\dagger, \widehat{a}_{\sigma',k}^\dagger\right] = \left[\widehat{a}_{\sigma,j}, \widehat{a}_{\sigma',k}\right] = 0. \tag{A.6}$$

Using this expansion, the many-body Hamiltonian can be rewritten as

$$\widehat{H} = \sum_{\sigma \in \{A,B\}}\left[\sum_{i,j} h_{ij}^\sigma \widehat{a}_{\sigma,i}^\dagger \widehat{a}_{\sigma,j} + \frac{1}{2}\sum_{ijk\ell} W_{ijk\ell}^\sigma \widehat{a}_{\sigma,i}^\dagger \widehat{a}_{\sigma,j}^\dagger \widehat{a}_{\sigma,k} \widehat{a}_{\sigma,\ell}\right] + \sum_{ijk\ell} W_{ijk\ell}^{AB} \widehat{a}_{A,i}^\dagger \widehat{a}_{B,j}^\dagger \widehat{a}_{B,k} \widehat{a}_{A,\ell}, \tag{A.7}$$

where

$$h_{ij}^{\sigma} = \int \phi_{\sigma,i}^*(x)\widehat{H}_{sp}^{\sigma}\phi_{\sigma,j}(x)dx\,, \tag{A.8}$$

are the one-body integrals, and

$$W_{ijk\ell}^{\sigma} = \iint \phi_{\sigma,i}^*(x_1)\phi_{\sigma,j}^*(x_2)\widehat{W}^{\sigma}(x_1,x_2)\phi_{\sigma,k}(x_1)\phi_{\sigma,\ell}(x_2)dx_1 dx_2\,, \tag{A.9}$$

$$W_{ijk\ell}^{AB} = \iint \phi_{A,i}^*(x_1)\phi_{B,k}^*(x_2)\widehat{W}^{AB}(x_1,x_2)\phi_{B,k}(x_2)\phi_{A,\ell}(x_1)dx_1 dx_2\,, \tag{A.10}$$

are the two-body interaction integrals. $\widehat{W}^{\sigma}(x_1^{\sigma},x_2^{\sigma})$ and $\widehat{W}^{AB}(x^A,x^B)$ denote the intra-component two-body interactions of components $\sigma$ and the inter-component interactions between two bosons in components $\sigma$ and $\sigma'$, respectively, while

$$\widehat{H}_{sp}^{\sigma} = \frac{\widehat{p}}{2m_{\sigma}} + \widehat{V}_{\text{ext}}^{\sigma}(x) \tag{A.11}$$

is the single-particle Hamiltonian of component $\sigma$. It is often useful to choose the eigenfunctions of the SP Hamiltonian (A.11) as the SP basis $|\phi_{\sigma,i}\rangle$. These will depend on the particular trapping potential $\widehat{V}_{\text{ext}}^{\sigma}(x)$ and can be simply obtained by employing an appropriate discrete variable representation (DVR) [65, 66] or finite difference (FD) method to solve the time-independent SP Schrödinger equation (A.11).

The matrix elements of the many-body Hamiltonian Eq.(A.7) can now be evaluated with respect to the many-body Fock-space defined by the SP eigenstates $\{|F_{\mu}\rangle\}$

$$|F_{\mu}\rangle = |n_1^A n_2^A \ldots n_i^A \ldots\rangle \otimes |n_1^B n_2^B \ldots n_i^B \ldots\rangle\,, \tag{A.12}$$

where $n_i^{\sigma}$ denotes the occupation number of component $\sigma$ in the state $|\phi_{\sigma,i}\rangle$. The Hamiltonian Eq.(A.7) preserves individual particle numbers, and we only consider the restricted Fock-space with definite particle numbers $N_{\sigma}$ (i.e., 2+2 or 3+3), that is, the occupation numbers take values between 0 and $N_{\sigma}$ and obey the constraint

$$\sum_i n_i^{\sigma} = N_{\sigma}\,. \tag{A.13}$$

Solving the many-body Hamiltonian in the full Fock-basis defined by the complete set of SP eigenstates is equivalent to solving the full Hamiltonian, but to treat the problem numerically we must truncate the basis. Such a truncation is the essence of any numerical approximation of a continuum many-body problem, and we consider the eigenvalue problem in the truncated Fock basis, i.e.

$$\sum_{\mu,\nu}\langle F_{\nu}|\widehat{H}|F_{\mu}\rangle\langle F_{\mu}|m\rangle|F_{\nu}\rangle = E^m\sum_{\nu}\langle F_{\nu}|m\rangle|F_{\nu}\rangle\,, \tag{A.14}$$

where $E^m$ and $|m\rangle$ are the $m$-th eigenvalue and eigenvector, respectively. This procedure is usually called Exact Diagonalization (ED) and has been widely used for Bose-Bose mixtures [67–73]. The standard truncation scheme is to only consider the $M_{\sigma}$ lowest energy eigenstates with respect to the SP Hamiltonian, which results in the dimension of the total truncated Hilbert space for bosons being given as

$$D = \prod_{\sigma\in\{A,B\}}\binom{N_{\sigma}+M_{\sigma}-1}{N_{\sigma}}\,. \tag{A.15}$$

The dimension of the truncated Hilbert space grows exponentially with the number of SP eigenstates, $M_{\sigma}$. This leads to highly expensive computational costs for achieving sufficiently

accurate results [67, 68] and therefore limits both the number of bosons and the number of excited states that can be accurately obtained at a given computational cost.

The following briefly presents the improved Exact Diagonalization (ED) scheme for Bose-Bose mixtures confined harmonically with two-body $\delta$-function interactions that we employ. This scheme allows us to compute numerically exact eigenstates with arbitrary inter- and intra-component coupling strengths and larger particle numbers for a much smaller truncated Hilbert space, making the calculations presented in this paper feasible at reasonable computational costs.

- Rather than truncating the many-body basis at a certain number of SP states $M_\sigma$ determined by their energy with respect to the SP Hamiltonian, we truncate the Fock-states with respect to the energy of the non-interacting many-body Hamiltonian. Therefore, we only consider Fock states whose energy is smaller than a specified value, $E_{max}$ as discussed in Refs [42, 43]. By increasing the value of $E_{max}$, the low-lying eigenstates of the many-body Hamiltonian can be accurately obtained even in a strongly interacting regime, avoiding the exponential growth of the Hilbert space obtained from the standard truncation method (equation (A.15)).

- Due to the symmetry of the many-boson wavefunction under permutations of any two bosons, only the Fock states with even parity contribute to the many-body eigenstates; therefore, the odd-parity Fock states can be removed safely. This not only significantly reduces the dimension of the truncated Hilbert space by nearly half, but also allows one to reach highly-excited bosonic eigenstates with fewer computational requirements. It should be noted that employing the even-odd parity of Fock states allows us to decompose the total Hilbert space into two subspaces with different symmetries, even- and odd-parity subspaces. In our case, we only focus on the even-parity Hilbert subspace since we are working on bosonic systems.

- We also use the so-called effective interaction approach, which replaces the two-body interaction with an effective interaction that incorporates information about the exact two-body solution. This is essentially a transformation of the interaction potential, which leads to exact two-body results for a limited number of eigenstates in the computational basis and which accelerates the convergence to the exact results of the many-body system in the truncated Hilbert space. For a more detailed description of the effective interaction approach for identical fermions confined harmonically, see Refs. [44–46]. For our system, where the bosons have the same mass and are trapped in a one-dimensional harmonic potential and the two-body interaction potential is modeled by the $\delta$-function, the effective interaction approach can be applied very efficiently. The two-body solutions are known analytically [41] and the Brody-Tamil-Moshinsky expansion [74] can be used to effectively evaluate the two-body interaction integrals (A.9) and (A.10) in terms of the relative coordinate wavefunctions.

A version of this computational method utilizing the effective interaction and the many-body energy truncation was also utilized to calculate the dynamics of five bosons in the previous work of some of the authors [75]. Although the idea of employing the energy-truncated Hilbert space and the effective interaction for obtaining the inter-component integrals $W^{AB}_{ijk\ell}$ has also been used in [76], we remark that in our improved Exact Diagonalization scheme we extend this by utilizing the effective interaction for both intra- and inter-components and take the symmetry of the many-body Fock basis into account. It is also interesting to note that recently another improved Exact Diagonalization scheme for ultra-cold atoms confined harmonically has also been introduced [77].

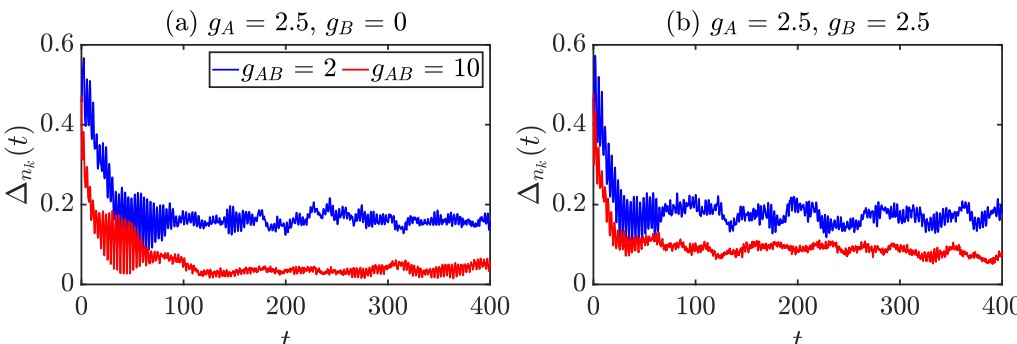

Figure 8: (a-b) Time evolution of $\Delta_{n_k}(t)$ for various intra-component coupling strengths $g_B$ and inter-component coupling strength $g_{AB}$. In all panels, the intra-component interaction of component A is fixed at $g_A = 2.5$.

## B   Momentum distribution function of the entire system

In addition to the density distribution function of component B and the trapping potential energy of the entire system, we also investigate another observable, namely the momentum distribution function of the entire system, $n(k, t)$, with the same initial state as given by Eq. (9) and the same energy window. We evaluate

$$\Delta_{n_k}(t) = \int |n(k,t) - n_{\mathrm{ME}}(k)| \, dk, \tag{B.1}$$

where $k$ denotes the momentum. As can be seen in Fig. 8, it is apparent that the behavior of the momentum distribution for the full system is consistent with the density distribution function of component B and the trapping potential energy of the full system. That is, it does not thermalize for small interactions $g_{AB} = 2$, while for larger $g_{AB} = 10$, it thermalizes when $g_A \neq g_B$, but not for $g_A = g_B = 2.5$. This confirms the emergence of quantum chaos in harmonically trapped Bose-Bose mixtures in 1D space with respect to the intra- and inter-component coupling strengths as predicted by the measures presented in the main text.

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
