# Peer review of "Quantum chaos in interacting Bose-Bose mixtures"

_SciPost Physics, doi:SciPost Phys. 15, 048 (2023)_

## Round 1 · Referee Report · Anonymous (Referee 1) · 2023-3-21

Strengths

1- This manuscript constitutes a very thorough investigation of the integrability-chaos transition in an empirically relevant quantum system: a two-component one-dimensional Bose gas

2- Authors diligently and properly used all the standard measures of quantum ergodicity. This paper could be used in a reading material for a class on quantum nonequilibrium.

Weaknesses

1- Unless I am mistaken, authors overlooked another integrable point: $g_{A} = g_{B} = g_{AB}$ (see e.g. [Y.-Q. Li, S.-J. Gu, Z.-J. Ying, and U. Eckern, Exact results of the ground state and excitation properties of a two-component interacting Bose system, Europhys. Lett., 61, 368 (2003).]). Curiously, no corresponding dip on the diagonal, at either 2a-c or 3a-c is visible. Yet, I think it would be important to plot the level spacing statistics for, say, 3+3 , $g_{A} = g_{B} = g_{AB} = 10$ and assess if it is "more Poissonian" than $g_{A} = g_{B} = 10, g_{AB} = 1.5$ and $g_{A} = g_{B} = 10, g_{AB} = 20$ .

2- It should be noted that the one-dimensional systems particles of the same mass can not fully serve as paradigmes of short-range-interacting gases. Unlike in two and three spatial dimensions where thermailzation occurs in a few two-body collision per particle [H. Wu, C.J. Foot, Direct simulation of evaporative cooling, J. Phys. B 29, L321 (1996).], in one dimension thermalization times are purely quantum entities, and thermalization takes longer than a few collisions. One-dimensional systems can be made more generic by introducing mass defects ( see e.g. [Z. Hwang, F. Cao, and M. Olshanii, Traces of Integrability in Relaxation of One-Dimensional Two-Mass Mixtures, J. Stat. Phys. 161, 467 (2015); Dmitry Yampolsky, N. L. Harshman, Vanja Dunjko, Zaijong Hwang, Maxim Olshanii, Quantum Chirikov criterion: Two particles in a box as a toy model for a quantum gas, SciPost Phys. 12, 035 (2022).] ). I would suggest comparing the relaxation times of Figs. 5-7 with the time scale associated with the two-body collisions: $\tau_{\text{2B}} \sim 1/(n v)$, where $n$ is the number density and $v \sim \sqrt{\epsilon /m}$, with $\epsilon$ being the total energy per particle.

Report

This is an important, rigorous, and thorough paper that deserves to be published. It will serve as a good reference text on quantum ergodicity.

Both of my comments are suggestions for improvement, and as such, they are optional.

Requested changes

There is no requested changes.

Optional changes:

1- Add a level spacing statistics histogram for $g_{A} = g_{B} = g_{AB} = 10$ and compare it to the $g_{A} = g_{B} = 10, g_{AB} = 1.5$ and $g_{A} = g_{B} = 10, g_{AB} = 20$ histograms;

2- Comment on an integrable point $g_{A} = g_{B} = g_{AB}$, regardless of whether it is visible in the Brody and kurtosis data or not.

3- Compare the relaxation times of Figs. 5-7 with the time scale associated with the two-body collisions. Cite [H. Wu, C.J. Foot, Direct simulation of evaporative cooling, J. Phys. B 29, L321 (1996); Z. Hwang, F. Cao, and M. Olshanii, Traces of Integrability in Relaxation of One-Dimensional Two-Mass Mixtures, J. Stat. Phys. 161, 467 (2015); Dmitry Yampolsky, N. L. Harshman, Vanja Dunjko, Zaijong Hwang, Maxim Olshanii, Quantum Chirikov criterion: Two particles in a box as a toy model for a quantum gas, SciPost Phys. 12, 035 (2022).]]

  • validity: top
  • significance: high
  • originality: top
  • clarity: top
  • formatting: perfect
  • grammar: perfect

Author:  Duong Anh-Tai Tran  on 2023-05-15  [id 3670]

(in reply to Report 1 on 2023-03-21)

We would like to thank the referees for reviewing our manuscript and for finding it a thorough investigation of chaos in few-body systems. In the following we address the comments of the referees and note the changes made to manuscript.

  • This is an important, rigorous, and thorough paper that deserves to be published. It will serve as a good reference text on quantum ergodicity. Our reply: We thank the referee for their time and interest into our work, and for their positive report.

  • Unless I am mistaken, authors overlooked another integrable point: $g_A=g_B=g_{AB}$ (see e.g. [Y.-Q. Li, S.-J. Gu, Z.-J. Ying, and U. Eckern, Exact results of the ground state and excitation properties of a two-component interacting Bose system, Europhys. Lett., 61, 368 (2003).]). Curiously, no corresponding dip on the diagonal, at either 2a-c or 3a-c is visible. Yet, I think it would be important to plot the level spacing statistics for, say 3+3, $g_A=g_B=g_{AB}=10$ and assess if it is "more Poissonian" than $g_A=g_B=10,g_{AB}=1.5$ and $g_A=g_B=10,g_{AB}=20$. Optional change: Comment on an integrable point $g_A=g_B=g_{AB}$, regardless of whether it is visible in the Brody and kurtosis data or not. Our reply: We thank the referee for pointing this out and below we show the level spacing distributions for $g_A=g_B=g_{AB}=10$ and compare it to the cases shown in the manuscript $g_A=g_B=10,g_{AB}=1.5$ and $g_A=g_B=10,g_{AB}=20$. The distributions show that there is no obvious difference between the $g_A=g_B=g_{AB}=10$ case and those with $g_{AB}\neq 10$. This is also confirmed by Fig.~2(a-c), Fig.~3(a-c) in the manuscript, with the point of equal interactions $g_A=g_B=g_{AB}$ not exhibiting any noticeable distinct features. Furthermore, we have re-calculated the distributions of $U_{mn}$ as shown in Fig. 4 for unequal interactions $g_{A}\neq g_{AB}$ and the results are qualitatively similar to those presented in the manuscript (Please see the attached file for the comparison). While the references provided by the referee suggest that the point $g_A=g_B=g_{AB}$ is integrable, this is only true for free space, not for harmonically trapped systems which we are considering. In the introduction of the model we noted that "the harmonic trap breaks integrability, and the system can thermalize, albeit on long timescales owing to the proximity of the system to integrability [K. F. Thomas, M. J. Davis and K. V. Kheruntsyan, Thermalization of a quantum Newton’s cradle in a one-dimensional quasi condensate, Phys. Rev. A 103, 023315 (2021)]". Our final conclusion is that the point $g_A=g_B=g_{AB}$ in our model is not integrable although it is special for having SU(2) symmetry. To make this more clear we have added a brief discussion of this to the text.

  • It should be noted that the one-dimensional systems particles of the same mass can not fully serve as paradigmes of short-range-interacting gases. Unlike in two and three spatial dimensions where thermailzation occurs in a few two-body collision per particle [H. Wu, C.J. Foot, Direct simulation of evaporative cooling, J. Phys. B 29, L321 (1996).], in one dimension thermalization times are purely quantum entities, and thermalization takes longer than a few collisions. One-dimensional systems can be made more generic by introducing mass defects ( see e.g. [Z. Hwang, F. Cao, and M. Olshanii, Traces of Integrability in Relaxation of One-Dimensional Two-Mass Mixtures, J. Stat. Phys. 161, 467 (2015); Dmitry Yampolsky, N. L. Harshman, Vanja Dunjko, Zaijong Hwang, Maxim Olshanii, Quantum Chirikov criterion: Two particles in a box as a toy model for a quantum gas, SciPost Phys. 12, 035 (2022).] ). I would suggest comparing the relaxation times of Figs. 5-7 with the time scale associated with the two-body collisions. Our reply: We thank the referee for pointing out the relation between the thermalization time and the two-body collision time in one-dimensional systems. We have checked and found that the thermalization time of the dynamics shown in Figs.~5,6,8 ($t_{rel}\sim 100$) is approximately two orders of magnitude larger than the two-body collision time ($t_{col}\sim 1$). In the revised version of our manuscript, we have mentioned the comparison and cited the suggested references.

Changes to the manuscript:

  • In second paragraph in Introduction on page 2, we added a new reference about chaos in bosonic mixtures [34].

  • In section 3.2.1 on page 8 we added: "We also note that in contrast to interacting two-component systems in free space [60], the trapped system we consider does not contain an integrable point at $g_A = g_B = g_{AB}$ . While the system does possess SU(2) symmetry at this point, there is nothing evident in the energy spacing statistics or the kurtosis that differentiates it from any other point along the diagonal $g_A = g_B$."

  • In section 3.2.2 on page 13 we added the sentence: "In addition, we note that the thermalization time of the dynamics shown in Figs. 5 and 6 is approximately two orders of magnitude larger than the two-body collision time [61–63]."

  • At the end of Appendix A on page 18 we added the sentence: "Although the idea of employing the energy-truncated Hilbert space and the effective interaction for obtaining the inter-component integrals $W^{AB}_{ijk\ell}$ has also been used in [76], we remark that in our improved Exact Diagonalization scheme, we extend this by utilizing the effective interaction for both intra- and inter-components and take the symmetry of the many-body Fock basis into account."

Attachment:

---

## Round 1 · Referee Report · Anonymous (Referee 2) · 2023-4-25

Strengths

  • This is a very thorough paper that addresses a lot of the natural questions involving thermalization of few-boson systems.

  • The authors find a number of interesting points where the system does not immediately develop chaotic level statistics.

Weaknesses

  • In some sense all the phenomena addressed in this paper are finite-size crossovers. It is clearly true that in a finite-size system that is nearly integrable, there will be regimes where the level statistics is not properly chaotic. The overall motivation for characterizing these crossovers is a bit opaque to me.

  • It is not clear to me why fitting to the Brody distribution adds any insight.

Report

On the whole I think this is a solid numerical paper that will be interesting to specialists in the field.
  • validity: high
  • significance: good
  • originality: good
  • clarity: good
  • formatting: perfect
  • grammar: excellent

Author:  Duong Anh-Tai Tran  on 2023-05-15  [id 3669]

(in reply to Report 2 on 2023-04-25)

We would like to thank the referees for reviewing our manuscript and for finding it a thorough investigation of chaos in few-body systems. In the following we address the comments of the referees and note the changes made to manuscript.

  • On the whole I think this is a solid numerical paper that will be interesting to specialists in the field. Our reply: We thank the referee for their time and interest into our work, and for their positive report.
  • In some sense all the phenomena addressed in this paper are finite-size crossovers. It is clearly true that in a finite-size system that is nearly integrable, there will be regimes where the level statistics is not properly chaotic. The overall motivation for characterizing these crossovers is a bit opaque to me. Our reply: The referee is correct that the results shown in the work can mostly be attributed to finite size effects which are of course inherent in such small systems. However, this does not imply that these are not interesting regimes to study. Motivated by advances in trapping on control of single [Endres \textit{et al}, Science \textbf{354} 1024 (2016)] and few-body systems [Bayha \textit{et al}, Nature 587, 583–587 (2020)] there are many recent works exploring the emergence of chaos in small systems [Zisling \textit{et al}, SciPost Phys. \textbf{10}, 088 (2021), Wittmann \textit{et al}, Physical Review E \textbf{105}, 034204 (2022), Yampolsky \textit{et al}, SciPost Phys. \textbf{12}, 035 (2022)], generally finding that $3$ or more particles are needed along with a broken symmetry induced by unequal masses or external potentials. In this work we go in a different direction and focus on inducing chaos through interaction effects in experimentally attainable two-component systems.

  • It is not clear to me why fitting to the Brody distribution adds any insight. Our reply: To study the emergence of chaos in these Bose-Bose mixtures we firstly rely on the energy spacing statistics which is a well-known indicator of chaos. While examples are shown in the manuscript it is not possible to show these distributions for the entire parameter space. Therefore, we use a fitting to the Brody distribution which allows to concisely differentiate between regions of Wigner-Dyson and Poissonian statistics. We believe this fitting succinctly highlights the chaotic and integrable regions of the parameter space which are then studied more rigorously in the following sections.

Changes to the manuscript:

  • In second paragraph in Introduction on page 2, we added a new reference about chaos in bosonic mixtures [34].

  • In section 3.2.1 on page 8 we added: "We also note that in contrast to interacting two-component systems in free space [60], the trapped system we consider does not contain an integrable point at $g_A = g_B = g_{AB}$ . While the system does possess SU(2) symmetry at this point, there is nothing evident in the energy spacing statistics or the kurtosis that differentiates it from any other point along the diagonal $g_A = g_B$."

  • In section 3.2.2 on page 13 we added the sentence: "In addition, we note that the thermalization time of the dynamics shown in Figs. 5 and 6 is approximately two orders of magnitude larger than the two-body collision time [61–63]."

  • At the end of Appendix A on page 18 we added the sentence: "Although the idea of employing the energy-truncated Hilbert space and the effective interaction for obtaining the inter-component integrals $W^{AB}_{ijk\ell}$ has also been used in [76], we remark that in our improved Exact Diagonalization scheme, we extend this by utilizing the effective interaction for both intra- and inter-components and take the symmetry of the many-body Fock basis into account."

---

## Round 2 · List of Changes

Changes to the manuscript:

  • In second paragraph in Introduction on page 2, we added a new reference about chaos in bosonic mixtures [34].

  • In section 3.2.1 on page 8 we added: "We also note that in contrast to interacting two-component systems in free space [60], the trapped system we consider does not contain an integrable point at $g_A = g_B = g_{AB}$ . While the system does possess SU(2) symmetry at this point, there is nothing evident in the energy spacing statistics or the kurtosis that differentiates it from any other point along the diagonal $g_A = g_B$."

  • In section 3.2.2 on page 13 we added the sentence: "In addition, we note that the thermalization time of the dynamics shown in Figs. 5 and 6 is approximately two orders of magnitude larger than the two-body collision time [61–63]."

  • At the end of Appendix A on page 18 we added the sentence: "Although the idea of employing the energy-truncated Hilbert space and the effective interaction for obtaining the inter-component integrals $W^{AB}_{ijk\ell}$ has also been used in [76], we remark that in our improved Exact Diagonalization scheme, we extend this by utilizing the effective interaction for both intra- and inter-components and take the symmetry of the many-body Fock basis into account."

---

## Editorial Decision

published